# Distinct Profiles of Desensitization of µ-Opioid Receptors Caused by Remifentanil or Fentanyl: In Vitro Assay with Cells and Three-Dimensional Structural Analyses

**DOI:** 10.3390/ijms24098369

**Published:** 2023-05-06

**Authors:** Eiko Uezono, Yusuke Mizobuchi, Kanako Miyano, Katsuya Ohbuchi, Hiroaki Murata, Akane Komatsu, Sei Manabe, Miki Nonaka, Takatsugu Hirokawa, Keisuke Yamaguchi, Masako Iseki, Yasuhito Uezono, Masakazu Hayashida, Izumi Kawagoe

**Affiliations:** 1Department of Anesthesiology and Pain Medicine, Juntendo University Graduate School of Medicine, Tokyo 113-8421, Japan; eiyu0825@gmail.com (E.U.);; 2Department of Pain Control Research, The Jikei University School of Medicine, Tokyo 105-8461, Japan; 3Department of Anesthesiology and Resuscitology, Okayama University Graduate School of Medicine, Dentistry and Pharmaceutical Sciences, Okayama 700-0194, Japan; 4Division of Cancer Pathophysiology, National Cancer Center Research Institute, Tokyo 104-0045, Japan; 5Department of Dentistry, National Cancer Center Hospital, Tokyo 104-0045, Japan; 6Tsumura Research Laboratories, Tsumura and Co., Ibaraki 300-1192, Japan; 7Department of Anesthesiology and Intensive Care Medicine, Nagasaki University Graduate School of Biomedical Sciences, Nagasaki 852-8501, Japan; 8Chemical Biology and In Silico Drug Design, Division of Biomedical Science, Faculty of Medicine, University of Tsukuba, Tsukuba 305-8575, Japan; 9Department of Anesthesiology and Pain Medicine, Juntendo Tokyo Koto Geriatric Medical Center, Tokyo 136-0075, Japan; 10Department of Pain Medicine, Juntendo University Graduate School of Medicine, Tokyo 113-8421, Japan; 11Supportive and Palliative Care Research Support Office, National Cancer Center Hospital East, Chiba 277-8577, Japan

**Keywords:** remifentanil, fentanyl, opioid receptor, desensitization, in silico simulation

## Abstract

Remifentanil (REM) and fentanyl (FEN) are commonly used analgesics that act by activating a µ-opioid receptor (MOR). Although optimal concentrations of REM can be easily maintained during surgery, it is sometimes switched to FEN for optimal pain regulation. However, standards for this switching protocol remain unclear. Opioid anesthetic efficacy is decided in part by MOR desensitization; thus, in this study, we investigated the desensitization profiles of REM and FEN to MOR. The efficacy and potency during the 1st administration of REM or FEN in activating the MOR were almost equal. Similarly, in β arrestin recruitment, which determines desensitization processes, they showed no significant differences. In contrast, the 2nd administration of FEN resulted in a stronger MOR desensitization potency than that of REM, whereas REM showed a higher internalization potency than FEN. These results suggest that different β arrestin-mediated signaling caused by FEN or REM led to their distinct desensitization and internalization processes. Our three-dimensional analysis, with in silico binding of REM and FEN to MOR models, highlighted that REM and FEN bound to similar but distinct sites of MOR and led to distinct β arrestin-mediated profiles, suggesting that distinct binding profiles to MOR may alter β arrestin activity, which accounts for MOR desensitization and internalization.

## 1. Introduction

Remifentanil (REM) and fentanyl (FEN) are commonly used narcotic analgesics [1,2,3] that activate opioid receptors (OR) during the perioperative and postoperative periods. ORs are classified into three subtypes: μ (MOR), δ (DOR), and κ (KOR) receptors, and their analgesic effects are mainly mediated by the MOR [4,5]. ORs belong to the G protein-coupled receptor (GPCR) family, which conjugate with Gi/o proteins and reduce intracellular cyclic adenosine monophosphate (cAMP) concentrations, activate K^+^ channels, and inhibit Ca^2+^ channels [4,6,7,8]. Cellular activities mediated by ORs are transduced through two signaling pathways: the G protein-mediated pathway mainly involved in analgesia and the β arrestin-mediated pathway related to adverse effects [7]; however, critical distinctions of these pathways are debatable [9,10]. β arrestin-mediated signaling is responsible for GPCR desensitization by inducing G protein α subunit dissociation from the activated GPCRs and mediating internalization of the activated GPCRs into cytosol, causing receptor degradation [5,6,11]. These β arrestin-induced events are associated with opioid tolerance [6,12,13].

REM is simple to manage in order to optimize its concentration during operation [1,3]; however, its activity is brief. Thus, we generally need to switch to FEN after operation [14,15,16]. However, there is currently no clear standard procedure for switching [12,13,14,17]. We occasionally experience inappropriate analgesic effects of REM or FEN during operation [2,12,13,17], which are attributed to opioid tolerance that is partly due to MOR desensitization [6,11,18,19]. In such clinical situations, opioid concentrations are increased or opioid switching from REM to FEN is usually considered [12,13]. To choose appropriate opioid switching, the mechanisms of regulation of MOR activity by REM and FEN should be clarified; however, few studies have done this. Furthermore, there is no consensus in the guidance [17], and such a switching approach remains ambiguous [12,13].

β arrestin-mediated signaling is involved in the control of GPCR desensitization and internalization processes, as for several protein kinases, such as G protein-regulated kinases [6,11,20]. Molecular mechanisms of the FEN-induced β arrestin signaling pathway have been clarified using an in silico simulation analysis with three-dimensional (3-D) MOR structure data [21]. Recently, the cryo-EM structure of human MOR complexed with FEN was reported [22], and the binding mode of REM to MOR, as well as that of FEN, was estimated by 3-D simulation and MOR mutation analyses [10,22]. The literature elucidates how FEN and FEN-related compounds activate G protein- and β arrestin-mediated signaling [10,22], which is useful when considering the signal transduction produced by FEN and REM in our present study. To establish the basic evidence for MOR desensitization and internalization, as well as desirable opioid switching methods between REM and FEN, we analyzed the MOR desensitization mechanism using an in vitro assay system using cells stably expressing MOR, performed a simulation study of 3-D MOR model analyses, and compared the MOR activity profiles. This study demonstrates that FEN has a stronger efficacy for MOR desensitization, whereas REM has superior effects on the internalization processes, both of which were mediated by β arrestin-dependent signals. Our 3-D simulation results further clarified an opioid switching theory from REM to FEN.

## 2. Results

### 2.1. Dose-Response Curves of REM, FEN, and D-Ala(2)-N-Me-Phe(4)-Glyol(5)-Enkephalin (DAMGO) with Cells Stably Expressing MOR

Figure 1A depicts the concentration-response curves obtained using the CellKey^TM^ assay for the activities of MOR in cells stably expressing MOR treated with varying concentrations of REM, FEN, and DAMGO. REM and FEN increased cellular impedance in a concentration-dependent manner, with their EC_50_ and E_max_ having almost similar values. Similarly, in the cAMP and β arrestin assay, REM and FEN exhibited similar potency and efficacy (Figure 1B,C).

### 2.2. Repetitive Administration of REM and FEN in Cells for Determination of MOR Desensitization with the CellKey^TM^

Repetitive administration of REM and FEN for the evaluation of MOR desensitization was conducted in cells stably expressing MOR, as previously reported in our laboratory in a protocol (Figure 2A) [23]. Cells were treated with 10^−10^ to 10^−8^ M REM or FEN (1st administration) and then further treated with the same concentrations of analgesics (2nd administration). REM and FEN exhibited desensitizing responses when administered repetitively (Figure 2B,C). The results showed that decreasing activity of REM occurred from 3 × 10^−9^ M, whereas FEN activity decreased from 10^−9^ M (Figure 2B,C). The ratio of the 2nd response to the 1st response exhibited a significant desensitization trend for FEN, suggesting that FEN has a stronger MOR desensitization ability than REM. Treatment with 10^−9^ M REM or FEN resulted in the most significant difference, as shown in Figure 2D.

### 2.3. Requirements of Secondary-Administrated Analgesic Concentrations to Achieve Equivalent Responses Compared with the 1st Response

To determine the concentrations of REM that elicited an equivalent response to that of FEN by the 2nd administration, varying concentrations of REM (10^−9^, 3 × 10^−9^, and 10^−8^ M) were administered. When 10^−9^ M REM was used in the 1st administration, REM at 3 × 10^−9^ M by the 2nd administration exhibited an equivalent response (Figure 3A). Furthermore, when 3 × 10^−9^ M REM was administered, 10^−8^ M REM exhibited an equivalent response (Figure 3B). In addition, when 10^−8^ M REM was administered, 10^−6^ M REM was required to achieve equivalent or greater responses (Figure 3C). Similarly, the concentrations of the 2nd administrated FEN that elicited equivalent responses were determined. As depicted in Figure 3D–F, 10^−8^ M of the 2nd administered FEN exhibited equivalency with the 1st administered 10^−9^ M FEN. When 3 × 10^−9^ M or 10^−8^ M FEN was used for the 1st administration, 10^−7^ M or 10^−6^ M FEN was required, respectively, to achieve equivalent responses (Figure 3D–F). These results suggest that 3- to 10-fold higher concentrations of the 2nd administrated REM were required to achieve equivalent 1st responses, whereas 10 to 100 folds of FEN concentrations were required to achieve equivalent 1st responses (Figure 3).

### 2.4. Opioid Switching Assay

We then determined the concentrations of the 2nd administered REM or FEN that achieved equivalent responses to those of the 1st administered analgesics, as previously reported with REM and morphine [23]. In these cases, we switched opioids from REM to FEN and vice versa (Figure 4). The 2nd administration of FEN (10^−9^ M) resulted in a response equivalent to that of the 1st administered REM at 3 × 10^−9^ M (Figure 4A). In contrast, none of the 2nd administrations of REM, even at 3 × 10^−8^ M (10-fold higher than the 1st concentration of FEN), reached equivalent responses to those of the 1st administered FEN at 3 × 10^−9^ M (Figure 4B).

### 2.5. Visual Internalization Assay of ORs in Cells Stably Expressing Halotag^®^-Tagged ORs

We performed an internalization assay of MORs as one of the β arrestin-mediated signaling pathway components [6,13,18,24]. In human embryonic kidney 293 (HEK293) cells stably expressing Halotag^®^MOR/pGlosensor^TM^-22F plasmid (pGS22F), the Halotag^®^ previously stained with the pH sensor ligand was used to visualize the internalized MOR induced by each compound. The pH sensor Halotag^®^ ligand is impermeable to cell membranes and binds to Halotag^®^-bound MOR on the membranes. The sensor showed increased red fluorescence as pH decreased, as previously reported [16,25]. Because internalized receptors are incorporated into the endoplasmic reticulum, which has low pH conditions, only internalized receptors can be visualized as red fluorescence over time [16,25]. For the MOR, the number and intensity of red spots were increased by REM administration compared with those of FEN administration over time (Figure 5A–C), and significant differences between REM and FEN were observed for 10^−8^ M and 10^−7^ M analgesics after 60 min of treatment (Figure 5C–E).

### 2.6. Binding Mode Estimation by Ligand Docking and Molecular Dynamics (MD) Simulation

We conducted an in silico simulation study [22,26,27,28] to assess the interaction of FEN and REM with the MOR (Figure 6A). The FEN-bound human MOR structure was used, and a docking model of REM was constructed. The MD simulation was then performed for 2000 ns (10 runs × 200 ns). Accordingly, a root-mean-square fluctuation (RMSF) analysis was performed to estimate the effect of each ligand on the MOR structure during the MD simulation (Figure 6B). REM exhibited strong fluctuations in the C-terminal helix of the MOR during simulation (Figure 6C). Regarding intracellular loops, a difference between FEN and REM was observed in the intracellular region 1 (ICL1) of the MOR. Figure 6C shows the amino acid residues, which present the difference over 0.5 Å (shown in green in Figure 6B).

## 3. Discussion

In the present study, we have demonstrated that REM and FEN, analgesics used during perioperative and postoperative management, have almost the same potency to activate MOR using CellKey^TM^, cAMP, and β arrestin assays. Although both analgesics elicited MOR desensitization when repeatedly administered (comparison of the 2nd to the 1st responses in the present study), FEN had stronger desensitization potency than did REM. When considering opioid switching, REM to FEN switching might require a lower dose of FEN to maintain equivalent MOR activities within the range of clinical concentrations (10^−9^ to 10^−8^ M) (Figure 4). Conversely, when FEN was switched to REM, the 2nd administration of REM no longer showed the same potency as that of the 1st-administered FEN even at 30-fold higher concentrations (Figure 4). Based on these findings, commonly using opioid switching during perioperative pain management from REM to FEM, or switching to FEN for patients in whom REM is not effective during an operation, seems to be an appropriate and reasonable method in clinical practice in terms of our present MOR desensitization profiles by REM or FEN.

We performed four independent assays, CellKey^TM^, cAMP, β arrestin, and MOR internalization, as previously reported [16,23,25]. The CellKey^TM^ assay system is used to measure most excitability proteins, including GPCRs, in real-time [8,29]. These changes reflect ligand-induced receptor reactions in whole-cell events, but it may be difficult to identify whether the responses are from the G protein- or β arrestin-mediated pathways. To further clarify these effects, we employed a cAMP assay that solely detects responses via G protein-mediated pathways [23,25]. The EC_50_ and E_max_ values between REM and FEN, measured using Cellkey^TM^, cAMP, and β arrestin assays, indicated similar potency and efficacy, demonstrating that REM and FEN similarly activate both G protein- and β arrestin-mediated pathways when first administrated to cells. Distinct MOR activity profiles between REM and FEN were observed in MOR desensitization and internalization assays. FEN had a stronger desensitization potency than did REM, whereas REM showed stronger internalization than did FEN. The findings provide novel perspectives on β arrestin recruitment characteristics in MOR signaling pathways [22,24,27]. One hypothesis indicates that, in the two-conformation style of β arrestin by activation of MOR, core conformation of β arrestin may proceed desensitization, while the tail conformation of β arrestin accelerates the internalization process, as indicated below.

Our in silico simulation analysis demonstrated that FEN and REM bound to different sites in the intracellular region of the MOR, particularly in the ICL1 and C-terminal helix 8 regions. The phenylethyl moiety of FEN stably interacts with a hydrophobic pocket of residues TM2–TM3 [22]. In contrast, the interaction between the methyl ester group in REM and the pocket was less stable in our MD simulation. This would lead to the differences in TM2 and ICL1, which may cause different β arrestin binding to MOR. Additionally, REM showed strong fluctuation against the helix 8 region; the region is important for tailed-β arrestin activation and receptor internalization [22,27]. These results suggest that FEN and REM differ in their activation mechanism for β arrestin, possibly by different receptor-ligand binding statuses, in spite of similar total responses of β arrestin activities in our study. Further investigation is necessary to elucidate the relationship between these ligand-induced structural changes and phenotypes caused by β arrestin-mediated signal activation.

A limitation of the present study is that we did not directly investigate the changes in the MOR core or tail conformational status induced by REM or FEN. However, we analyzed their effects, such as desensitization (or internalization states), led by β arrestins bound to the core or tail site, respectively. Further detailed analyses of β arrestin activity mediated by REM or FEN should be performed with the help of precise comparison of three-dimensional structural analysis of MOR by REM or FEN stimulation.

Overall, the present results suggest that when switching from REM to FEN in clinical practice, lower concentrations of FEN may be sufficient to maintain analgesic efficacy. The present practice of switching in and after surgery seems to be appropriate and reasonable. The different activities of REM and FEN may be due to their binding sites on the MOR. Future studies should be performed to investigate how much these properties, along with β arrestin binding kinetics, contribute to ligand-mediated signaling differences of MOR.

## 4. Materials and Methods

### 4.1. Construction of Plasmids and Generation of Stable Cell Line

The construction of plasmids and generation of stable cell lines for the MOR have been previously described [23,25]. HEK293 cells (ATCC^®^, Manassas, VA, USA), and HEK293 cells stably expressing Halotag^®^MOR, with or without pGS22F, were generated by transfection of the constructed plasmids using Lipofectamine reagent (Life Technologies Corp., Carlsbad, CA, USA). These were selected based on OR activity measured using the CellKey^TM^ assay or the Glosensor^®^ cAMP assay as previously reported [16,23,25].

### 4.2. Cell Culture

HEK293 cells stably expressing Halotag^®^MOR/pGS22F were cultured in Dulbecco’s modified Eagle’s medium supplemented with 10% fetal bovine serum albumin, penicillin (100 U/mL), streptomycin (100 mg/mL), 5 μg/mL puromycin (InvivoGen, San Diego, CA, USA), and 100 μg/mL hygromycin (FUJIFILM Wako Pure Chemical Corporation, Osaka, Japan) for Halotag^®^MOR/pGS22F as previously reported [16,23,25].

### 4.3. Chemicals

The following reagents were used: DAMGO (D-Ala(2)-N-Me-Phe(4)-Glyol(5)-enkephalin), forskolin, IBMX (3-isobutyl-1-methylxanthine), Ro 20-1724 (Sigma-Aldrich, Saint Louis, MO, USA), HaloTag^®^ pH Sensor Ligand (Promega, Madison, WI, USA), Hoechst 33342 (Dojinkagaku, Kumamoto, Japan), morphine hydrochloride (Takeda Pharmaceutical Co., Ltd., Tokyo, Japan), and FEN and REM (Janssen Pharmaceutical K. K., Tokyo, Japan). Forskolin, IBMX, Ro 20-1724, and all inhibitors were diluted with dimethyl sulfoxide, whereas the other reagents were diluted with H_2_O.

### 4.4. Functional Analysis of ORs with the CellKey^TM^ System

The CellKey^TM^ assay system has been previously described [16,23,25]. Briefly, cells stably expressing Halotag^®^MOR/pGS22F were seeded at densities of 6.0 × 10^4^ in poly-D-lysine (Sigma-Aldrich, Saint Louis, MO, USA)-coated CellKey^TM^ 96-well microplates and incubated for 24 h. The wells were washed with a CellKey^TM^ buffer composed of Hank’s balanced salt solution (in mM: 1.3 CaCl_2_·2H_2_O, 0.81 MgSO_4_, 5.4 KCl, 0.44 KH_2_PO_4_, 4.2 NaHCO_3_, 136.9 NaCl, 0.34 Na_2_HPO_4_, and 5.6 D-glucose) containing 20 mM 4-(2-hydroxyethyl)-1-piperazineethanesulfonic acid (HEPES) and 0.1% bovine serum albumin (BSA). The cells were incubated for 30 min at 28 °C before the assay was conducted in accordance with a protocol described previously [16,23,25]. Changes in the impedance current (Ziec) in each well were measured at 10 s intervals for up to 30 min while considering the first 5 min as baseline, and Ziec measurements were obtained for 25 min after administration of each compound. The Ziec values for each sample were corrected using the values obtained for the negative control samples. The positive control in ΔZiec measurements included cells treated with DAMGO for Halotag^®^MOR/pGS22F, as reported previously [25].

### 4.5. Functional Analysis of ORs Using Repetitive Administration Protocol with CellKey^TM^

Cells were incubated for 30 min at 28 °C before the assay, in accordance with a previously described protocol [23]. Changes in Ziec in each well were measured at 10 s intervals for up to 30 min while considering the first 5 min as baseline, and Ziec measurements were obtained for 25 min after administration of each compound. The second measurement was performed after incubation for 30 min at 28 °C again (Figure 2A). In some cases, several inhibitors were administered to the cells as previously reported [23] if necessary during this incubation period. Changes in Ziec in each well were measured at 10 s intervals for up to 30 min while considering the first 5 min as a baseline, and Ziec measurements were obtained for 25 min after administration of each compound in the same way.

### 4.6. Intracellular cAMP Assay with GloSensor^®^

The GloSensor^®^ cAMP assay was performed as described previously [16,23,25]. Briefly, cAMP accumulation was analyzed using cells stably expressing Halotag^®^MOR/pGS22F. The cells were seeded at 4.0 × 10^4^ cells/well in 96-well clear-bottom plates (Corning Inc., Corning, NY, USA) and incubated for 24 h. After washing the cells with CellKey^TM^ buffer without BSA, the cells were equilibrated with diluted GloSensor^®^ reagent at 25 ± 2 °C room temperature for 2 h, and the baseline luminescence intensity was measured for 15 min. After the baseline measurement, cells were treated with the test compounds for 10 min, and forskolin (3 × 10^−6^ M) was then added. The luminescence intensity was measured every 2.5 min for 30 min using Synergy^TM^ H1 (BioTek Instruments Inc., Winooski, VT, USA), time-luminescence curves were traced, and the area under the curve (AUC) values of time-luminescence intensity were calculated. The responses to each compound were expressed by subtracting the AUC of each compound from that of the negative control sample (treatment with forskolin alone). Data are presented as the percentage of intracellular cAMP inhibition, calculated by dividing the corrected AUC by that of the standard sample. The standard sample included cells treated with DAMGO (at 10^−5^ M) for Halotag^®^MOR/pGS22F.

### 4.7. β Arrestin Recruitment Assay with PathHunter^®^

The β arrestin recruitment assay was performed according to the protocol for PathHunter^®^ (DiscoverX, Fremont, CA, USA) and as previously reported [16,23]. U2OS OPRM1 cells were seeded at a density of 1.0 × 10^4^ cells/well in 96-well clear-bottom white plates and incubated for 48 h. The cells were stimulated for 90 min (MOR) with a dilution series of ORs at 37 °C and 5% CO_2_, and PathHunter^®^ working detection solution was added. Luminescence intensity was measured using Flex Station 3 (Molecular Devices, San Jose, CA, USA) for 1 h at 25 ± 2 °C room temperature. Data are expressed as the percentage of the positive control (maximum signal intensity of each test compound/maximum signal intensity of positive control [16,23]).

### 4.8. Internalization Assay of MOR

MOR internalization assay was performed according to a previously described method [16,25]. Briefly, cells stably expressing Halotag^®^MOR/pGS22F (1.2 × 10^5^ cells/well) were seeded in a polyethylenimine-coated 8-well chamber slide system. After incubation for 24 h, the cells were washed once with the internalization buffer (10 mM HEPES, 140 mM NaCl, 5 mM KCl, 2 mM CaCl_2_, 1 mM MgCl_2_, and 10 D-glucose at pH 7.4) and stained with Hoechst 33342 for 10 min followed by the pH sensor ligand (0.5 × 10^−6^ M) for 15 min (incubated in 5% CO_2_ at 37 °C). Red spots in the cells were recorded every 30 min after compound treatment for 180 min, using an FV10i confocal laser scanning microscope (Olympus, Tokyo, Japan). Both REM and FEN were administered immediately before the start of observation. The numbers of both red spots and nuclei were counted using MetaMorph^®^ 7.7 (Molecular Devices, San Jose, CA, USA). Data were quantified as “sum of intensity/cell” and normalized to that of cells before compound administration (% of the sum of intensity/cell before compound administration).

### 4.9. Ligand Docking and Molecular Dynamics Simulation

The human MOR complex with FEN was obtained from the Protein Data Bank (8EF5) [22]. Energy minimization of REM and FEN was performed using the OPLS3e force field in LigPrep in Maestro (Schrödinger LLC, NY, USA). The MOR structure [28] was prepared for docking simulations using the Protein Preparation Wizard Script with Maestro. REM docking simulation was performed using the Glide SP docking program (Schrödinger LLC, NY, USA). The hydrogen-bonding constraint between the side-chain COO-group of Asp147 was introduced because this hydrogen-bond formation is conserved in known complexes of Class A GPCRs bound to a variety of agonists and antagonists. The positions for further analysis were manually selected based on the docking score and the preceding database [22].

We conducted MD simulations to assess the differences in binding modes between FEN and REM. The docking structures were subjected to three independent MD simulations with different initial velocities for each target using Desmond v2.3. The OPLS3e force field was used for the simulations. The initial model structures were placed in a large POPC bilayer and TIP3P water molecules with periodic boundary conditions using an orthorhombic 10 Å layer simulation box. The system was neutralized, and an ionic force of 0.15 M was set by adding Na^+^ and Cl^−^ ions. After minimization and relaxation of the model, the MD production phase was performed for 200 ns with a time step of 2 fs in an isothermal–isobaric (NPT) ensemble at 300 K and 1 bar using a Langevin thermostat. The long-range electrostatic interactions were computed using the Smooth Particle Mesh Ewald method [22]. The MD trajectories were saved every 10 ps for analysis. All system setups were performed using the Maestro software 2020-2 (Schrödinger, LLC, NY, USA).

The obtained trajectory was processed using the AMBER11 ptraj tool to calculate the RMSF. Figures showing MOR structures were prepared using PyMol 2.4 (Schrödinger, LLC, NY, USA).

### 4.10. Statistical Analysis

Data analysis, concentration-response curve fitting, and fitting smoothing curve using a locally weighted scatterplot smoothing (Lowess) method were performed using GraphPad Prism 8 (GraphPad Software, La Jolla, CA, USA). Data are presented as means with a standard error of the mean (SEM) for at least three independent experiments. Statistical analyses were performed using the Mann–Whitney test or two-way analysis of variance (ANOVA), followed by the Tukey–Kramer test (GraphPad Prism 8). Statistical significance was set at *p* < 0.05.

## 5. Conclusions

The present results suggest that when switching from REM to FEN, lower concentrations of FEN may be sufficient to maintain analgesic efficacy, and switching from REM to FEN seems reasonable based on their desensitization profiles. The mechanisms of the different desensitization and internalization properties of REM and FEN could be due to their binding sites and subsequent changes in the 3-D MOR structure, particularly in the regions of TM6 and TM7. The present results shown might be helpful for the establishment of guidelines for opioid switching, in the aspect of basic scientific research.

## Figures and Tables

**Figure 1 ijms-24-08369-f001:**
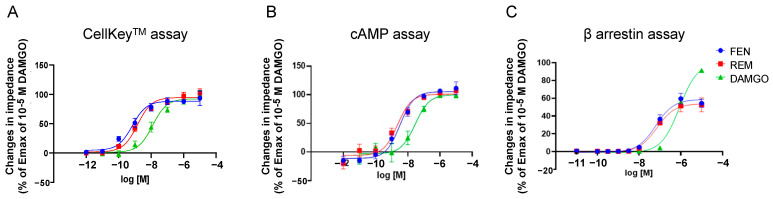
Concentration-response curves of remifentanil (REM), fentanyl (FEN), and DAMGO with cells stably expressing µ opioid receptors (MORs) with CellKeyTM, cAMP, and β arrestin assays. Cells expressing MOR were treated with REM, FEN, or DAMGO (10^−12^ to 10^−5^ M), and changes in impedance (ΔZiec) were measured. Concentration-response curves were prepared by calculating ΔZiec relative to the data obtained for the positive control (10^−5^ M DAMGO) with CellKey^TM^ (**A**). Intracellular cAMP levels were measured using the GloSensor^®^ cAMP assay. Data were prepared by calculating cAMP levels relative to the data obtained with 10^−5^ M DAMGO and are presented as mean ± SEM for three independent experiments (*n* = 6) (**B**). β arrestin assay was performed in the cells treated with each of the compounds (10^−11^ to 10^−5^ M), and data were prepared by calculating intracellular β arrestin levels relative to the data obtained for the positive control (10^−5^ M DAMGO) (**C**). All data are presented as mean ± SEM for three independent experiments (*n* = 6).

**Figure 2 ijms-24-08369-f002:**
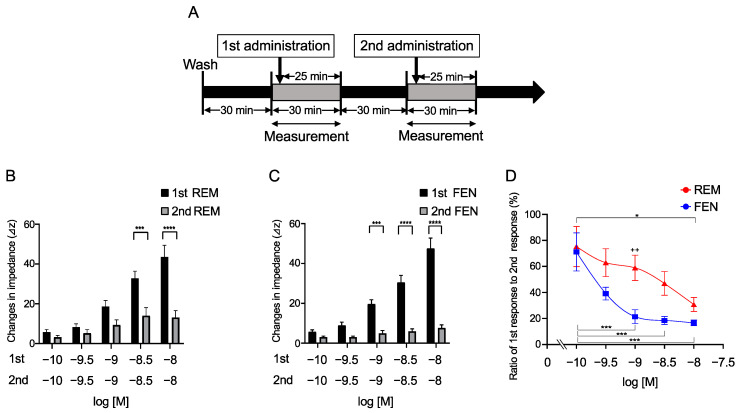
Repetitive administration of REM or FEM in the cells expressing MOR with CellKeyTM. Cells expressing MOR were treated with REM or FEN twice for each indicated concentration (10^−10^ to 10^−8^ M). Changes in impedance (ΔZiec) were measured with the CellKey^TM^ (**A**). REM was repetitively administrated as shown in (**A**), and data are reported (**B**). FEN was also repetitively administrated (**A**), and data are reported (**C**). Summary of the concentration-response curves of REM or FEN is depicted in (**D**). Data are presented as mean ± SEM for three independent experiments (*n* = 6). Relative ratio of 2nd response to 1st response (%) is reported. * *p* < 0.05, *** *p* < 0.001, **** *p* < 0.0001 vs. 1st responses by REM or FEN. ++ *p* < 0.01 vs. data obtained with 10^−9^ M FEN.

**Figure 3 ijms-24-08369-f003:**
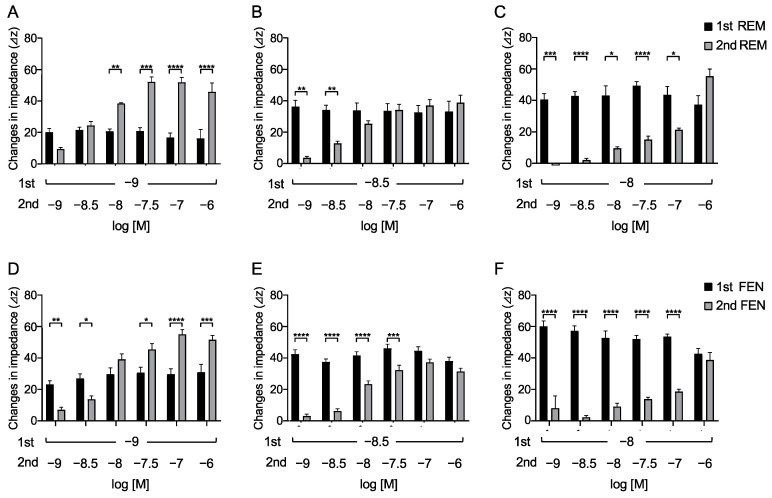
Overall analyses to find concentrations of secondary-administered REM or FEN to achieve responses equal to the 1st responses in cells expressing MOR with the CellKey^TM^. Cells expressing MOR were treated with varying concentrations of REM or FEN (10^−9^ to 10^−6^ M). After measuring changes in impedance (ΔZiec) by the 1st administration (10^−9^, 3 × 10^−9^, and 10^−8^ M) of analgesics, 2nd administration (10^−9^ to 10^−6^ M) was conducted, and changes in impedance were measured (ΔZiec) in cells. The 1st administration of REM at 10^−9^ M (**A**), 3 × 10^−9^ M (**B**), or 10^−8^ M (**C**) and the 1st administration of FEN at 10^−9^ M (**D**), 3 × 10^−9^ M (**E**), or 10^−8^ M (**F**) are independently depicted. * *p* < 0.05, ** *p* < 0.01, *** *p* < 0.001, **** *p* < 0.0001 vs. the 1st responses by FEN or REM.

**Figure 4 ijms-24-08369-f004:**
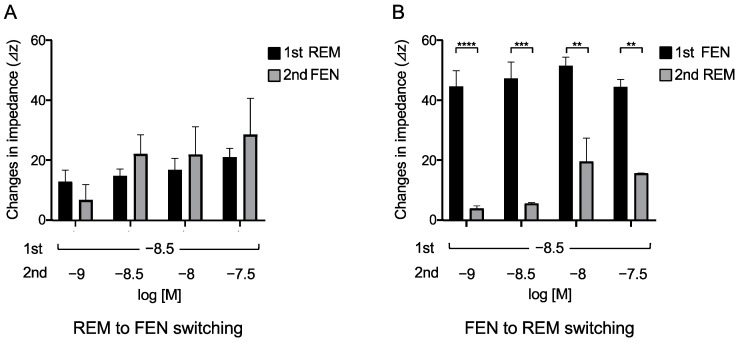
Opioid switching from REM to FEN and FEN to REM in cells expressing MOR assessed with CellKey^TM^. Cells expressing MOR were treated with REM or FEN at specific concentrations. In the protocol, analgesics of the 2nd administration were changed from REM to FEN (**A**) or FEN to REM (**B**), and the changes in impedance (ΔZiec) were measured. Data are presented as mean ± SEM for three independent experiments (*n* = 6). ** *p* < 0.01, *** *p* < 0.001, **** *p* < 0.0001 vs. 1st responses by REM or FEN.

**Figure 5 ijms-24-08369-f005:**
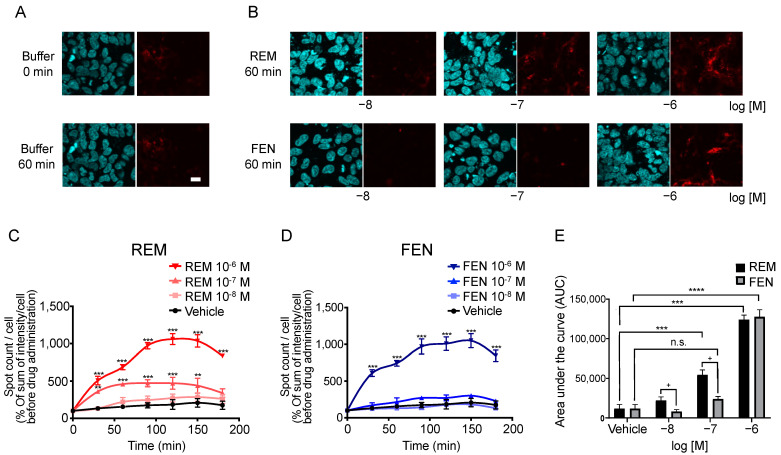
Internalization of MOR induced by REM or FEN in HEK293 cells stably expressing Halotag^®^MOR. HEK293 cells stably expressing Halotag^®^MOR were stained with Hoechst 33342 (blue) and the Halotag pH sensor ligand (red) and treated with REM, FEN, or vehicle for up to 180 min, images were taken after 60 min of incubation (**A**,**B**), and they were observed at the indicated time points (**C**–**E**). To quantify the internalization levels, the numbers and intensities of red spots/cell were counted using MetaMorph^®^ 7.7. Data were quantified by the sum of intensity/cell and normalized as % of sum of intensity/cell before REM or FEN administration (**C**–**E**). All data are presented as mean ± SEM (*n* = 3–5). Bar = 20 µm. ** *p* < 0.01,*** *p* < 0.001, **** *p* < 0.0001 vs. vehicle, n.s.: not significant vs. vehicle, + *p* < 0.05 vs. REM at the same concentration.

**Figure 6 ijms-24-08369-f006:**
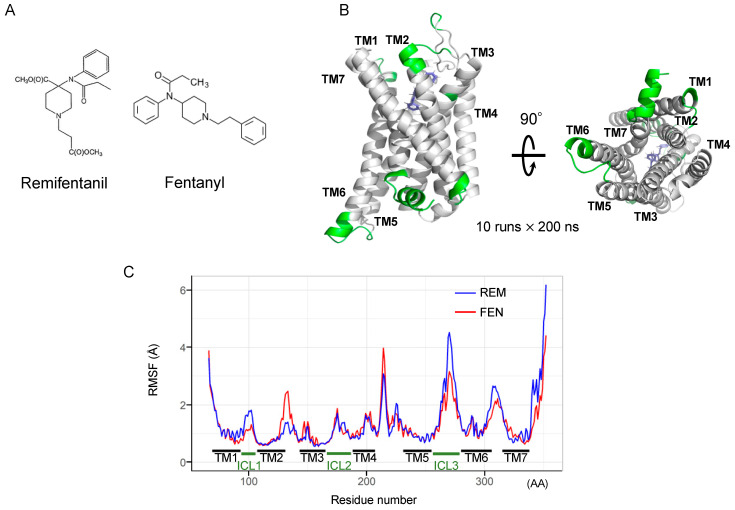
Determination of molecular dynamics (MD) simulation of REM- or FEN-bound MOR complex. Molecular structures of REM and FEN (**A**). Determination of MD simulation was conducted with REM- or FEN-bound human MOR. Fluctuation analysis during MD simulation revealed the difference in the influence of the ligand docking (**B**). The difference between REM (blue) and FEN (red) based on the fluctuation analysis is visualized (**C**). Green indicates the difference over 0.5 Å between the fluctuations of REM and FEN. TM: transmembrane, ICL: intracellular region, RMSF: root mean square fluctuation, AA: amino acids.

## Data Availability

Not applicable.

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
