# Peer review of "Distinct Profiles of Desensitization of µ-Opioid Receptors Caused by Remifentanil or Fentanyl: In Vitro Assay with Cells and Three-Dimensional Structural Analyses"

_ijms, 2023, doi:10.3390/ijms24098369_

Round 1

Reviewer 1 Report

In this paper, the authors compare the efficacy performance of difference REM-FEN treatment schedule, such as using REM first followed by REM, using FEN first followed by FEN, using REM first followed by FEN and using FEN first followed by REM. I have a few questions as follows.

Question 1: Could the author introduce more about the data they use for the analysis? In all the analysis, the authors do not provide any population (even though the unit / subject is cell) characteristic summary and do not adjust any potential confounding factors. The analysis results can be biased unless they can 100% sure that the population characteristics under each treatment schedule is almost the same. Or at least the authors can provide some subgroup analysis to ensure that that their findings are robust across different subgroups.

Question 2: Several analysis are quite straightforward. For example, in Figure 2D, 5C and 5D, the authors compare efficacy ratio over different nm values and cell counts over different time. The statistical analysis is based on scatter comparison. It would be a good try for the authors to fit smoothing non-parametric curve to see if the whole trend (overall trend of efficacy / cell counts over nm values / time by treating nm values / times as continuous variable) are the same or different.

The English quality is good and clear.

Author Response

Thank you for your comments which greatly improve our manuscript.

Response to Question 1:

In our in vitro experiments with the cells, we used established cell lines prepared ourselves and cells that are commercially available, and both cell lines have been used elsewhere and by us, and some of our papers with these cell lines were published (see ref. 16, 23 and 25).

In the present study, we used these two cell lines; these were 1) cells stably expressing Halotag®️MOR/pGS22F for CellKeyTM, cAMP, and internalization assays, and 2) another cell line was U2OS OPRM1 specially designed for β arrestin recruiting assay with a part of PathHunter®️ kit and they were purchased from DiscoverX, Fremont, CA, USA.

Halotag®️MOR/pGS22F cells were used from stocked cells (-80°C) and 2-8 passages of cell lines of Halotag®️MOR/pGS22F were used throughout the experiment in the manuscript and U2OS OPRM1 was used up with every one experiment. Also, all experiments were performed with 96-well plates and each of experiments (i.e., for producing concentration-response curves, data with FEN-FEN; REM-REM; FEN-REM; REM-FEN applied experiments using CellKeyTM), was done with a 96-well plate in almost similar experimental conditions; all experiments were conducted at air-conditioned 25 ± 2°C room temperature during daytime schedules.

Some of the information is added in the revised version.

Response to Question 2:

We asked our professional statistician at the Clinical Research Support Center at our university (The Jikei University School of Medicine). According to suggestions by the statistician, we tried to fit the smoothing curve with Lowess method in GraphPad Prism 8. Results of Figures 2D, 5C, and 5D with this analysis were redrawn and put revised revision.

Reviewer 2 Report

Summary

The study investigated the desensitization profiles of the analgesics Remifentanil (REM) and Fentanyl (FEN) to μ-opioid receptor (MOR). The efficacy and potency of the first administration of REM and FEN in activating MOR were similar, but the second administration of FEN resulted in stronger MOR desensitization potency than REM, whereas REM showed higher internalization potency than FEN. The study suggests that different β-arrestin-mediated signaling caused by FEN or REM leads to their distinct desensitization and internalization processes. Three-dimensional analysis with in silico binding of REM and FEN to MOR models highlighted that REM and FEN bound similar but distinct sites of MOR, suggesting that distinct binding profiles to MOR may alter β-arrestin activity that accounts for MOR desensitization and internalization. The results suggest that switching from REM to FEN seems reasonable based on their desensitization profiles, and lower concentrations of FEN may be sufficient to maintain analgesic efficacy. The study might be helpful for the establishment of guidelines for opioid switching from a basic scientific research perspective.

Opinions

The authors present an excellent study with well-designed experiments and a comprehensive analysis of the results. And I recommend publishing the manuscript once my questions below are answered (not necessarily need to add the answers to the manuscript). 

  1. Can you provide more details about the CellKeyTM assay system, including its sensitivity, specificity, and potential limitations? How have you validated the use of this assay for measuring OR activity?

  1. Can you explain why the GloSensor®️ cAMP assay was chosen as the method for analyzing cAMP accumulation? Have you compared the results from this assay to those obtained using other methods for measuring cAMP levels?

  2. How have you validated the use of the PathHunter®️ β-arrestin recruitment assay for measuring OR activity? What are the potential limitations of this assay, and how have you addressed these limitations in your study?

Author Response

Thank you for your comments and we are grateful for your comments and we would like to answer your questions, as follows.

Response to Opinion 1

CellKeyTM system we used in our study for the analysis of MOR activity are cell-based dielectric assay and is now widely employed as a whole-cell, label-free, and real-time assay, especially for GPCR assays (i.e., ref. 8 and 29 in this manuscript). Validation of the CellKeyTM system compared to other traditional assay methods such as cAMP and GTPrS assays were performed and this method is indeed statistically comparable to those of traditional GPCR assays (see also ref. 25 and also Kuroda Y, et al. Biomed Pharmacother, 141:111800, 2021).

Response to Opinion 2

We actually used to use several types of cAMP assay methods including cell-based ELISA cAMP measurement. For comparison of the GloSensor®️-based cAMP assay and a traditional ELISA cAMP assay, we found that the GloSensor®️-based cAMP assay was highly sensitive and also can be measured in a real-time mode that was not detectable in ELISA cAMP assay. Accordingly, we choose this assay as a highly-sensitive and detectable cAMP assay in a real-time mode.

Response to Opinion 3

For β arrestin assays, there are several types of assays; β arrestin recruiting assay with a visual qualitative method using GFP-tagged β arrestin, and spectrometry assay with illuminated β arrestin beneath plasma membranes upon stimulation. We, this time, used a commercially available β arrestin recruiting assay presented by DiscoverX in the USA as PathHunter®️. This includes the ready-to-use cell line where β arrestin and MOR were expressed and provided as a 96-well plate. It can be spectrometrically detectable with an apparatus such as Flex Station 3 (Molecular Devices, San Jose, CA, USA).

Reviewer 3 Report

Congratulations to the authors for a well conducted scientific paper as well as a theme. For someone who on a regular basis uses TCI with remifentanil and afterwards fentanil for postoperative analgesia ( you stated in abstract for “better” but I would say optimal because of remifentanil short acting mechanism of action) I find this theme very apealing.
Why you only investigate fentanil (my opioid of choise too) but sufentanil is gaining much more attention as well so please do incorporate as well.

Author Response

Thank you for your comments and excellent suggestion. In the revised abstract, we changed "better" to "optimal" according to your suggestion. As for the reviewer's proposal, it would be nice to include the results with sufentanil as another candidate for anesthetic during the perioperative and postoperative periods. However, in Japan, it still not be permitted as a clinically available anesthetic. It would be interesting to try such type of experiment with sufentanil if sufentanil is approved by the Pharmaceuticals and Medical Devices Agency in Japan.

Round 2

Reviewer 1 Report

Thanks so much for the revisions. The results are much improved.